# Rethinking Model Selection and Decoding for Keyphrase Generation with Pre-trained Sequence-to-Sequence Models

**Di Wu, Wasi Uddin Ahmad, Kai-Wei Chang**
University of California, Los Angeles
{diwu,kwchang}@cs.ucla.edu, wasiahmad@ucla.edu

## Abstract

Keyphrase Generation (KPG) is a longstanding task in NLP with widespread applications. The advent of sequence-to-sequence (seq2seq) pre-trained language models (PLMs) has ushered in a transformative era for KPG, yielding promising performance improvements. However, many design decisions remain unexplored and are often made arbitrarily. This paper undertakes a systematic analysis of the influence of model selection and decoding strategies on PLM-based KPG. We begin by elucidating why seq2seq PLMs are apt for KPG, anchored by an attention-driven hypothesis. We then establish that conventional wisdom for selecting seq2seq PLMs lacks depth: (1) merely increasing model size or performing task-specific adaptation is not parameter-efficient; (2) although combining in-domain pre-training with task adaptation benefits KPG, it does partially hinder generalization. Regarding decoding, we demonstrate that while greedy search achieves strong F1 scores, it lags in recall compared with sampling-based methods. Based on these insights, we propose DESEL, a likelihood-based decode-select algorithm for seq2seq PLMs. DESEL improves greedy search by an average of 4.7% semantic F1 across five datasets. Our collective findings pave the way for deeper future investigations into PLM-based KPG.

## 1 Introduction

Keyphrases encapsulate the core information of a document. Due to their high information density, they have been found valuable in areas such as information retrieval (Wu and Bolivar, 2008; Dave and Varma, 2010; Kim et al., 2013; Boudin et al., 2020), document clustering (Hammouda et al., 2005), summarization (Zhang et al., 2004), and text classification (Berend, 2011). A keyphrase is termed a *present keyphrase* if it is explicitly found within the document and an *absent keyphrase* otherwise. The task of identifying present keyphrases is defined as *keyphrase extraction* (KPE), whereas

*keyphrase generation* (KPG) involves predicting both types of keyphrases.

Recently, pre-trained language models (PLMs) have been widely incorporated in KPG (Chowdhury et al., 2022; Zhao et al., 2022) via sequence-to-sequence (seq2seq) generation, with promising performance on zero-shot (Kulkarni et al., 2022), multilingual (Gao et al., 2022), and low-resource (Wu et al., 2022a) KPG. However, existing literature typically focuses on a specific subset of important components in this pipeline, such as data construction and loss design, while making arbitrary choices for the others (Zhao et al., 2022; Ray Chowdhury et al., 2022; Wu et al., 2022a; Garg et al., 2022). As a result, KPG systems are often compared under different assumptions and the effect of the arbitrary design choices remains unclear. To bridge this gap, this paper focuses on two crucial questions that have not been systematically explored:

1. *Which PLM leads to the best KPG performance when fine-tuned?*
2. *What is the best decoding strategy?*

In practice, sub-optimal choices for these factors could lead to optimizing an unnecessarily large model or sub-optimal results decoded from a strong KPG model. To answer these two questions, we conduct in-depth analyses on KPG with (1) PLMs of diverse size and pre-training strategies and (2) a diverse set of decoding strategies.

To begin with, we posit that *seq2seq PLMs are inherently suitable to KPG* (§3). By drawing the correlations with a strong graph-based KPE algorithm, we show that these PLMs implicitly compute *phrase centrality* (Boudin, 2013) in their decoder attention patterns. This knowledge is also directly translatable to a strong ranking function for KPE. On the other hand, encoder-only models fail to carry such centrality information.

Next, we search for the best *seq2seq PLM* for KPG fine-tuning (§4). While common strategies for other NLP tasks might advocate for (1) scaling

up the model size, (2) in-domain pre-training, or (3) task adaptation, do these approaches hold the same merit for KPG? Our findings reveal that a singular emphasis on scaling or task adaptation does not ensure efficient performance improvement. In contrast, in-domain pre-training consistently bolsters performance across both keyphrase types and can benefit from task adaptation. A robustness analysis reveals that a proper model choice and data-oriented training approaches are complementary: without the latter, stronger PLMs are more vulnerable to perturbed input, with over 14% recall drop under named variation substitutions and over 5% recall drop under input paraphrasing.

Decoding strategy is also an essential component in PLM-based KPG, but much under-explored by current literature. In §5, we thoroughly compare six decoding strategies including greedy search, beam search, and sampling-based methods. Results suggest that when only generating a single sequence consisting of concatenated keyphrases, greedy search achieves a strong F1 score. However, aggregating the predictions from multiple sampled sequences outperforms greedy search due to a much higher recall.

Based on these findings, we introduce DeSel, a likelihood-based selection strategy that selects from sampled phrases to augment the greedy search predictions. DeSel utilizes the probability of phrases from greedy search's predictions as the baseline to filter out noisy predictions from a set of sampled keyphrase candidates. Experiments on five KPG datasets show that DeSel consistently improves greedy decoding by 7.9% $F1@M$ for present keyphrases, 25% $F1@M$ for absent keyphrases, and 4.7% Semantic F1 for all keyphrases, achieving state-of-the-art KPG performance, underscoring the importance of carefully examining the design choice of KPG.

To summarize, our primary contributions are:

1. An in-depth exploration of the intrinsic suitability of seq2seq PLMs for KPG.
2. A comprehensive examination of effective strategies for model enhancement in KPG, spotlighting the merits of specific combinations and their implications for robustness.
3. We establish the trade-off between accuracy and concept coverage for different decoding algorithms. Then, we introduce a probability-based decode-select mechanism DeSel that consistently improves over greedy search.

4. Our research illuminates the profound impact of under-emphasized factors on KPG performance. To facilitate future research on KPG, we release our code and models at `https://github.com/uclanlp/DeepKPG`.

## 2 Preliminaries

### 2.1 Keyphrase Generation

**Problem Definition**  We represent an example for KPG as a tuple $(\mathcal{X}, \mathcal{Y})$, corresponding to the input document $\mathcal{X} = (x_1, x_2, ..., x_d)$ and the set of human-written reference keyphrases $\mathcal{Y} = \{y_1, y_2, ..., y_n\}$. Following Meng et al. (2017), $y_i$ is classified as a *present keyphrase* if it is a substring of $\mathcal{X}$ or an *absent keyphrase* otherwise. The KPG task requires predicting $\mathcal{Y}$ in any order, and the KPE task only requires predicting present keyphrases (Turney, 2000).

**Evaluation**  We adopt lexical-based and semantic-based evaluation to evaluate a model's predictions $\mathcal{P} = \{p_1, p_2, ..., p_m\}$ against $\mathcal{Y}$. For lexical evaluation, we follow Chan et al. (2019) and use the $P@M$, $R@M$, and $F1@M$ scores. $\mathcal{P}$ and $\mathcal{Y}$ are stemmed with the Porter Stemmer (Porter, 1980) and the duplicates are removed before the score calculation. For semantic evaluation, we follow Wu et al. (2023) and report $SemP$, $SemR$, and $SemF1$. Note that lexical metrics are separately calculated for present and absent keyphrases while the semantic metrics are calculated with all the phrases. We repeat all the experiments with three random seeds and report the averaged performance.

**Benchmark**  Meng et al. (2017) introduce KP20k, which contains 500k Computer Science papers. Following their work, we train on KP20k and evaluate on the title and abstracts from the KP20k test set as well as four out-of-distribution testing datasets: Inspec (Hulth, 2003), Krapivin (Krapivin et al., 2009), NUS (Nguyen and Kan, 2007), and SemEval (Kim et al., 2010). Table 5 summarizes the statistics of all testing datasets.

**Baselines**  We consider two strong supervised encoder-decoder models from Ye et al. (2021):

1. **CopyTrans**: A Transformer (Vaswani et al., 2017) with copy mechanism (See et al., 2017).
2. The **SetTrans** model, which performs order-agnostic KPG. The model uses control codes trained via a k-step target assignment algorithm to generate keyphrases in parallel.

As the goal of this work is thoroughly studying PLM-based methods, we only provide the results of the strongest baselines as a point of reference. We also include other baselines in appendix section G. In our analysis, we also use MultipartiteRank (MPRank, Boudin (2018)), a performant graph-based unsupervised KPE algorithm. More details about MPRank are discussed in §3.1.

## 2.2 Sequence-to-Sequence PLMs

In this work, we focus on fine-tuning Transformer-based sequence-to-sequence PLMs **BART** (Lewis et al., 2020) and **T5** (Raffel et al., 2020) for KPG with the "One2Seq" formulation. Concretely, following Ye and Wang (2018) and Yuan et al. (2020), we use a separator token ; to join all the target keyphrases as the target sequence $\mathcal{Y} = (y_1 ; \ldots ; y_n)$. The models are trained with the cross-entropy loss for generating $\mathcal{Y}$ based on $\mathcal{X}$. At test time, greedy decoding is used, followed by a post-processing stage to segment the output sequence into individual phrases. We provide implementation details and hyperparameters in appendix D.

## 3 Do PLMs inherently carry significant knowledge of keyphrases?

Existing studies have justified their use of seq2seq PLMs by drawing the close relationship between the pre-training tasks of BART (denoising language modeling) or T5 (unified text-to-text transfer) and the formulation of KPG (Gao et al., 2022; Zhao et al., 2022; Wu et al., 2022a) or KPE (Kong et al., 2023). However, there is a lack of an in-depth understanding of *why* seq2seq PLMs should be chosen for keyphrase-related tasks. In this section, we reason based on *phrase centrality* (Litvak and Last, 2008; Boudin, 2013) and show that PLMs with autoregressive decoders, including seq2seq PLMs, carry attention heads that approximately function as centrality assigners and naturally as potent keyphrase rankers.

### 3.1 Centrality of Phrases

The concept of phrase centrality originated from graph-based KPE, where keyphrase candidates are represented as nodes. Various graph centrality measures are used to determine a phrase's importance in the document. We use MPRank in our analysis, which encodes closeness-based and eigenvector-based centrality (Boudin, 2013). MPRank first uses rules to obtain $C$ noun phrase candidates and then

| Model | Size | Head | $\rho$ | Head | $\tau$ |
|---|---|---|---|---|---|
| ***Encoder-only PLMs*** | | | | | |
| BERT-base | 110M | 3-0 | 0.300 | 3-0 | 0.206 |
| BERT-large | 340M | 0-5 | 0.351 | 0-6 | 0.246 |
| ***Decoder-only PLMs*** | | | | | |
| gpt2 | 117M | 0-11 | 0.626 | 0-11 | 0.479 |
| gpt2-medium | 345M | 1-6 | 0.630 | 1-6 | 0.480 |
| gpt2-large | 774M | 0-13 | 0.627 | 0-13 | 0.478 |
| gpt2-xl | 1.5B | 0-6 | 0.626 | 0-6 | 0.476 |
| ***Seq2seq PLMs*** | | | | | |
| BART-base | 140M | 0-6 | 0.608 | 0-6 | 0.459 |
| BART-large | 406M | 0-9 | 0.585 | 0-9 | 0.438 |
| T5-small | 60M | 4-4 | 0.624 | 4-4 | 0.471 |
| T5-base | 223M | 8-4 | 0.621 | 8-4 | 0.466 |
| T5-large | 770M | 0-2 | 0.628 | 0-2 | 0.471 |
| T5-3B | 3B | 0-8 | **0.648** | 0-8 | **0.494** |

Table 1: Correlation between keyphrase candidates' attention weights and centrality scores. $l$-$h$ denotes attention head $h$ in layer $l$. Both $h$ and $l$ start from index 0. We report the attention head that achieves the best scores. The highest score is boldfaced.

performs lexical clustering to group the candidates into topic clusters. Next, each candidate is represented as a graph node and connected with the candidates from other topic clusters. TextRank (Mihalcea and Tarau, 2004) is used to obtain a centrality score $c_i$ for each of the nodes $n_i$. We refer the readers to Boudin (2018) for further details.

### 3.2 Attention intensities in BART and T5 decoders encode phrase centrality

Using MPRank as a lens, we first investigate the extent to which PLMs implicitly represent centrality information. We use the paper titles and abstracts from the KP20k test set as the probing set. Each probing instance is fed into a PLM and the attention weights from the self-attention layers are collected. For the $h^{th}$ attention head at layer $l$, we denote the attention from token $i$ to token $j$ as $\alpha_{i \to j}^{l,h}$. For the $j^{th}$ token in the noun phrase candidate $n_i$, the global attention weight on it is

$$a_{ij}^{l,h} = \sum_{k=1,\ldots,L} \alpha_{k \to j}^{l,h}, \quad (1)$$

where $L$ is the length of the text after tokenization. Then, the attention weight of $n_i$ is calculated as

$$a_i^{l,h} = |n_i| \sum_j a_{ij}^{l,h}, \quad (2)$$

where $|n_i|$ denotes the number of tokens in $n_i$.

We study four families of models: BART, T5, BERT, and GPT-2 (Radford et al., 2019). For

BART and T5, we use their decoder attentions. We correlate $a_i^{l,h}$ with $c_i$ using Spearman correlation $\rho$ and Kendall's Tau $\tau$ and present the best correlation for each model in Table 1. Surprisingly, BART and T5 decoders contain *attention heads that encode phrase centrality similarly as MPRank*. The head with the best correlation generally appears in the lower layers, indicating that centrality understanding may be more related to low-level features. Also, the upper bound of correlation strength grows with model size for T5 while does not grow for BART. Beyond *centrality assigners*, these attention heads are also potent *keyphrase extractors*: simply ranking the noun phrase candidates by $a_i^{l,h}$ achieves similar $F1@5$ for present keyphrases or $SemF1@5$ score as MPRank (appendix B).

Evaluating other types of PLMs, we find that BERT's attention heads only show weak centrality knowledge, with only 0.246 best Kendall Tau with MPRank. On the other hand, GPT-2 exhibits a similar pattern to the decoders from seq2seq PLMs, indicating that the observed pattern is strongly associated with *autoregressive decoders*. As centrality is generally correlated with global importance, our result aligns with the observations that masked language modeling tends to exploit local dependency while causal language modeling can learn long-range dependencies (Clark et al., 2019; Vig and Belinkov, 2019).

In summary, through attention-based analyses, we reveal novel insights into the underlying keyphrase knowledge of PLMs with autoregressive decoders. Such knowledge can be employed explicitly (via ranking) or implicitly (via fine-tuning and prompting) to facilitate KPG. In the rest of the paper, we focus on rethinking two basic designs for KPG with seq2seq PLMs.

## 4 Influence of PLM Choice for KPG

Three crucial design options exist for using seq2seq PLMs for KPG: *the choice of PLM* to fine-tune, *the fine-tuning data and objective*, and *the decoding strategy*. Previous work focuses on fine-tuning objective and data construction (Meng et al., 2021; Ray Chowdhury et al., 2022; Garg et al., 2023) while often making the other two choices in an *ad hoc* way, making it difficult to compare among approaches. This section dives into the first question by evaluating three "conventional wisdoms":

1. Using PLMs with *more parameters* (§4.1).
2. Using *in-domain* PLMs (§4.2).

3. Using *task-adapted* PLMs (§4.3).

### 4.1 The scaling law for keyphrase generation

Although the effect of model sizes has been explored for a range of tasks, it is poorly understood in the KPG literature, where most recent works employ a single PLM with 100M to 500M parameters (Kulkarni et al., 2022; Wu et al., 2022b; Zhao et al., 2022). To establish a common ground, we measure the performance of fine-tuning BART-base/large (purple line) and T5-small/base/large/3B (green line) and report the results on KP20k in Figure 1.

Surprisingly, fine-tuning BART or T5 is extremely *parameter-inefficient* compared to task-specific architectures trained from scratch[1]. For instance, although T5's performance consistently increases with the model size, around 8x more parameters are required to achieve the same Present $F1@M$ on KP20k as SetTrans and 30x more parameters are required to have a better $SemF1$. Closer inspection shows that SetTrans excels in *recall* via its parallel control codes and the set loss. In comparison, limited by the learning formulation and decoding strategy, fine-tuned seq2seq PLMs fall behind in their recall of important keyphrases. In §5, we will show that this problem can be alleviated with a simple decode-then-select strategy.

**BART vs. T5**  BART and T5 display similar scaling for $F1@M$ and $SemF1$. However, compared to T5, BART's recall scores increase more readily than the precision scores. At the same number of parameters, BART also performs better on absent keyphrases. One possible reason is that BART's text infilling objective is more advantageous for learning the knowledge for constructing spans absent from text (Wu et al., 2022a).

**Which score is more sensitive to scaling?**  Compared to recall, *precision* is more sensitive to model size. For example, T5-small achieves 98% $SemR$ compared to the 50x larger T5-3B. In addition, *absent keyphrases* scores are more sensitive. Overall, this suggests that small models are able to extract relevant keyphrases, but learn to *selectively omit* unimportant keyphrases and *create more absent keyphrases* as the model size grows. Indeed, the average number of predicted keyphrases decreases from T5-small (6.75), T5-base (5.74), and T5-large (5.66), to T5-3B (5.48), while the number of absent keyphrases increases from T5-small (0.91),

---

[1]We note that this claim is orthogonal to the observations that PLMs are *data-efficient* (Wu et al., 2022a).

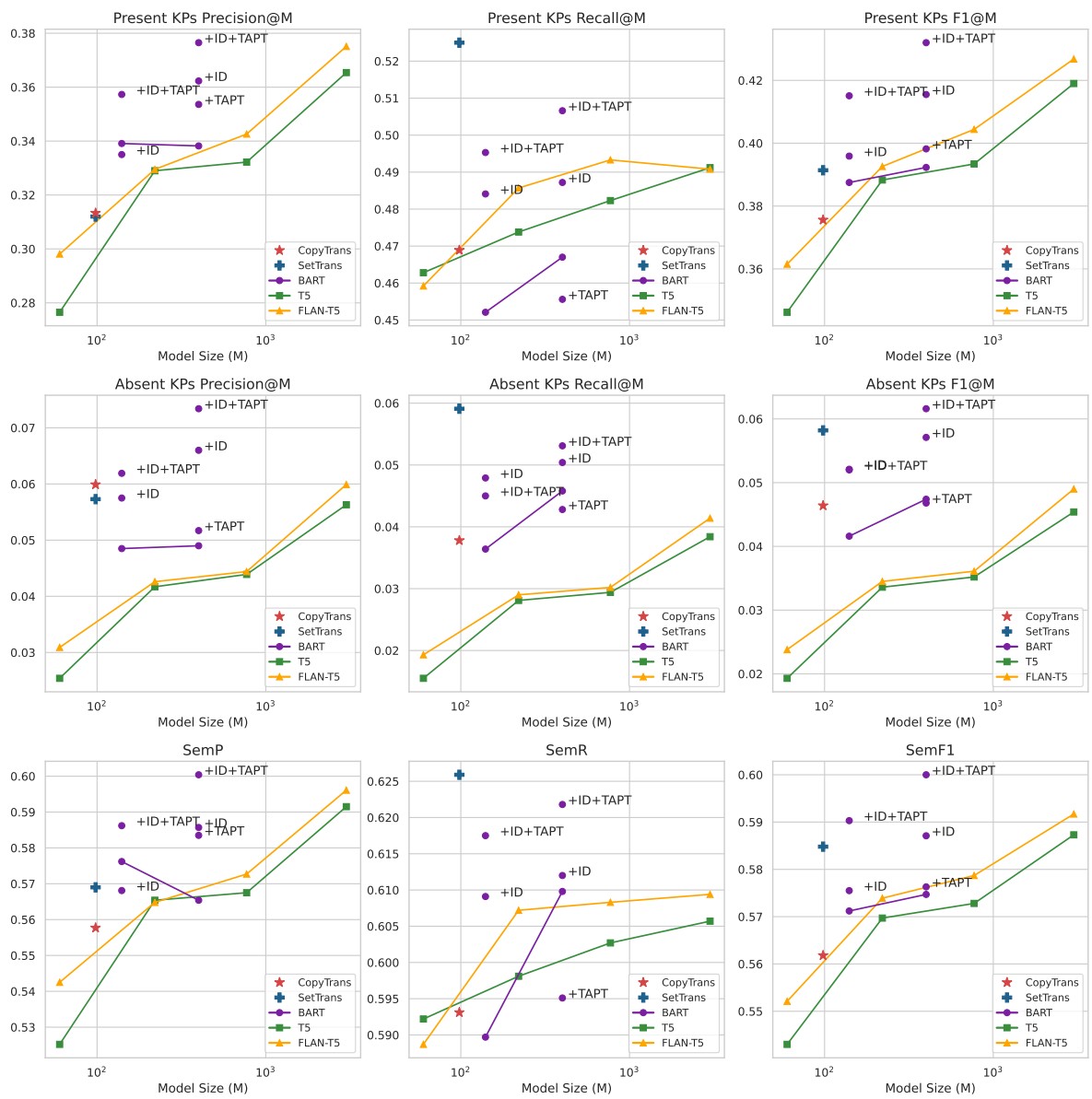

Figure 1: KP20k test performance of models of various sizes and pre-training strategies. +ID = using in-domain SciBART. +TAPT = second-stage training on OAGKX. The results are averaged over 3 random seeds.

T5-base (0.99), T5-large (1.01), to T5-3B (1.05).

## 4.2 Domain knowledge is crucial to accurate keyphrase generation

In-domain pre-training has been shown effective in a wide range of tasks requiring extensive domain knowledge (Beltagy et al., 2019; Lee et al., 2019). As keyphrases often contain domain-specific terminologies, we hypothesize that the domain of a PLM greatly affects its keyphrase generation ability. To test this hypothesis, we pre-train in-domain BART models SciBART-base and SciBART-large from scratch using the paper titles and abstracts from the S2ORC dataset (Lo et al., 2020). The processed dataset contains 171.7M documents or 15.4B tokens in total. The models are pre-trained on text infilling for 250k steps with batch size 2048, learning rate 3e-4, 10k warm-up steps, and polynomial learning rate decay. We present data processing and model training details in appendix C.

The results of fine-tuning SciBART are presented with "+ID" (for "In-Domain") in Figure 1. As expected, SciBART significantly improves over BART for all three F1 metrics, outperforming the much larger T5-3B. Notably, SciBART also has *better parameter efficiency* compared to general domain models: scaling from SciBART-base to SciBART-large provides a much larger growth in $SemF1$ compared to scaling up BART and T5.

## 4.3 Task-adaptive pre-training is more effective with in-domain models

Task-adaptive pre-training (TAPT) is another common technique for adding task-specific supervision signals (Gururangan et al., 2020). In this section, we analyze the effect on KPG performance of adding two types of TAPT stages to seq2seq PLMs: *keyphrase generation* or *instruction following*.

**Keyphrase Pre-training** We directly use Key-BART (Kulkarni et al., 2022) (denoted as "+TAPT" in Figure 1), which is trained using the OAGKX dataset (Çano and Bojar, 2020) on KPG with keyphrases corrupted from the input. To investigate the effects of TAPT on in-domain PLMs, we also fine-tune SciBART on OAGKX with batch size 256, learning rate 3e-5, and 250k steps. We denote this model as "+ID+TAPT" in Figure 1.

**Instruction Pre-training** Recently, instruction tuning has been introduced to improve the generalization ability of PLMs (Mishra et al., 2022; Ouyang et al., 2022). As KPG is relevant to classic NLP tasks such as information extraction and summarization, we hypothesize that training with instruction data also serves as TAPT for KPG[2]. To confirm, we benchmark FLAN-T5 (Chung et al., 2022), a family of T5 models fine-tuned on instruction following datasets (yellow line in Figure 1).

**TAPT struggles to improve absent KPG but is more effective with in-domain models.** Figure 1 suggests that both TAPT strategies lead to a similar amount of improvement in the present keyphrase F1@M and SemF1. Surprisingly, the absolute gain is small and TAPT hardly improves absent keyphrase performance. For KeyBART, although its pre-training data (OAGKX) has a similar percentage of absent keyphrases as KP20K (32% vs. 37%), its objective (recovering present keyphrases from corrupted input) might still be different from absent keyphrase generation. For FLAN-T5, we find that the KPG-related tasks in its pre-training tasks often contain very short input text, representing a significant distribution mismatch with KP20k. However, when applied on the in-domain SciBART, TAPT can greatly improve the performance on KP20k. Combined with §4.2, we conclude that *in-domain pre-training is more important for KPG and TAPT serves a complementary secondary role.*

---

[2]In fact, some variants of the keyphrase extraction task are included in popular instruction datasets such as NIv2 (Wang et al., 2022) and Alpaca (Taori et al., 2023).

| Model | Before | After | Δ |
|---|---|---|---|
| *Name Variation Substitution* | | | |
| BART-large | 0.476 | 0.410 | -0.066 (**13.9%↓**) |
| SciBART-large | **0.531** | 0.420 | -0.131 (20.9%↓) |
| KeyBART | 0.463 | 0.398 | **-0.065** (14.2%↓) |
| SciBART-large+TAPT | 0.525 | **0.434** | -0.091 (17.4%↓) |
| *Paraphrasing* | | | |
| BART-large | 0.463 | 0.453 | **-0.010** (**2.3%↓**) |
| SciBART-large | **0.524** | **0.494** | -0.030 (5.8%↓) |
| KeyBART | 0.466 | 0.407 | -0.059 (12.6%↓) |
| SciBART-large+TAPT | 0.522 | 0.487 | -0.035 (6.8%↓) |

Table 2: Results for the probing experiments. "Before" and "After" indicate the recall scores before and after perturbation. Δ denotes the score change from "Before" to "After". The best score is boldfaced.

## 4.4 Analysis: are strong KPG models sensitive to input perturbations?

As in-domain and task-adapted PLMs already greatly benefit KPG, are data augmentation techniques no longer necessary? In this section, we reveal that these designs increase the model's sensitivity to input perturbations, and data augmentation is still desired for better generalization.

### 4.4.1 Method

We design two input perturbations on KP20k to check the behaviors of BART-based KPG models.

**Name variation substitution** We construct 8905 perturbed inputs by replacing present keyphrases with their name variations linked by Chan et al. (2019). Ideally, a robust KPG model would have a similar recall for the original phrases and the name variations as they appear in the same context.

**Paraphrasing** We leverage `gpt-3.5-turbo` to paraphrase 1000 documents into a scientific writing style. A good KPG model is expected to retain similar recall for phrases that present both before and after paraphrasing since the inputs describe the same ideas. The detailed prompt and several examples are presented in appendix E.

### 4.4.2 Results

Table 2 presents the perturbation results. We observe that models often fail to predict name variations when they appear in place of the original synonyms, while successfully maintaining their predictions given paraphrased inputs. This indicates that the models may overfit to the high-frequency keyphrases in the training set and input augmentation methods such as Garg et al. (2023) are necessary to correct this pattern.

In addition, domain-specific or task-adapted models exhibit a larger performance drop compared to BART-large, suggesting a trade-off between domain/task specificity and generalization. Pre-trained on large-scale keyphrase data, Key-BART may rely more on syntax and position information in the data and thus be less sensitive to synonym change. On the other hand, pre-trained on a large-scale scientific corpus, SciBART is more robust than KeyBART on different scientific writing styles beyond the ones available in KP20k.

### 4.5 Discussion

We summarize the main conclusions derived from the empirical results presented in this section:

- Naively scaling up BART and T5 is parameter-inefficient on KP20k compared to SetTrans.

- Domain knowledge is crucial for KPG performance and improves parameter efficiency.

- Task-adaptive training with keyphrase or instruction tuning data only significantly improves KPG with in-domain models.

- In-domain pre-training and TAPT harm generalization in different ways and data augmentation during fine-tuning is desired.

## 5 Decoding Strategy for KPG

While it is well-known that decoding strategies can strongly affect text generation quality (Fan et al., 2018; Holtzman et al., 2020), surprisingly there has been little study about decoding PLM-based KPG models. Previous studies often directly use greedy search and variants of beam search (Gao et al., 2022; Zhao et al., 2022; Wu et al., 2022a), limiting the understanding of PLMs fine-tuned for KPG. To bridge this knowledge gap, we first carefully evaluate six decoding strategies on the strongest PLM-based KPG model. We then propose a simple yet effective *decode-select* strategy to mitigate the observed deficiencies of greedy search.

### 5.1 Multi-sequence decoding: the trade-off between coverage and quality

We focus on decoding the SciBART-large+TAPT model fine-tuned on KP20k, under the budget varying from 1 to 20 samples. The following six decoding algorithms are compared. For each algorithm, their hyperparameters are chosen based on the KP20k validation set.

1. *Greedy search*.
2. *Beam search*. We set the beam size to the number of desired samples.
3. *Diverse beam search* (Vijayakumar et al., 2018). We set the number of groups to the number of desired samples and the weight of the dissimilarity term to $\lambda_g = 0.1$.
4. *Vanilla sampling*. We further apply a temperature scaling $\tau = 0.7$.
5. *Top-k sampling* (Fan et al., 2018). We use temperature $\tau = 0.7$ and $k = 2$ as we find a large $k$ harms the generation quality.
6. *Nucleus sampling* (Holtzman et al., 2020). We set $p = 0.95$ and temperature $\tau = 0.5$.

Figure 2 presents the semantic-based evaluation results as a function of sample size. In the single sample setting, greedy search achieves a strong $SemF1$, only slightly outperformed by diverse beam search. For other methods, we observe a clear trade-off between their information coverage ($SemR$) and the noise in the final output ($SemP$) as the number of samples grows. Nevertheless, all these methods are able to outperform greedy search at a certain sample size, indicating that single-sequence decoding is sub-optimal.

### 5.2 A simple decode-select strategy boosts the performance of greedy decoding

Greedy search captures the correlations in human-written labels but suffers from *local decisions* and *path dependency*: high-quality keyphrases can be missed with improbable first tokens. However, naively outputting the union of multiple sampled sequences brings excessive noise. To achieve the balance between the two, we introduce DeSel, a simple and effective three-stage decoding strategy:

1. **De**code one sequence $G$ via greedy search.
2. Sample $n$ sequences $\{S_1, ..., S_n\}$ to collect a set of candidate keyphrases $S$.
3. **Sel**ect high quality phrases $\{s_1, ..., s_m\} \subset S$ and output the sequence $(G \,;\, s_1 \,;\, ... \,;\, s_m)$.

For step 3, we estimate $\Pr(s_i|\mathcal{X})$ for every phrase $s_i$ in the $n$ samples and $\Pr(g_j|\mathcal{X})$ for every phrase $g_j \in G$. Then, we use $G$ as a baseline to select at most $m$ most probable $s_i$ that satisfies

$$\Pr(s_i|\mathcal{X}) \geq \frac{\alpha}{|G|} \sum_{g_j \in G} \Pr(g_j|\mathcal{X}), \qquad (3)$$

where $\alpha$ is a hyperparameter controlling the trade-off between precision and recall. The probability estimation is obtained with either the original

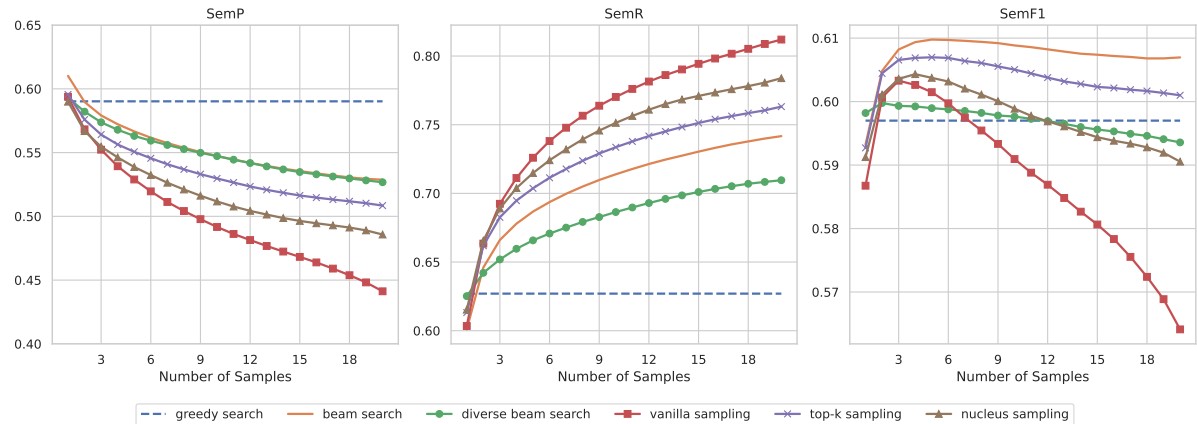

Figure 2: A comparison of six strategies for decoding from the SciBART+TAPT model. Greedy search achieves strong performance while performing worse than beam search and sampling with multiple samples.

| Method | KP20k | | | Inspec | | | Krapivin | | | NUS | | | SemEval | | |
|---|---|---|---|---|---|---|---|---|---|---|---|---|---|---|---|
| | $P$ | $A$ | $Sem$ | $P$ | $A$ | $Sem$ | $P$ | $A$ | $Sem$ | $P$ | $A$ | $Sem$ | $P$ | $A$ | $Sem$ |
| CopyTrans | 0.376 | 0.046 | 0.562 | 0.333 | 0.023 | 0.569 | 0.365 | 0.063 | 0.547 | 0.429 | 0.044 | 0.579 | 0.321 | 0.022 | 0.377 |
| SetTrans | 0.391 | 0.058 | 0.585 | 0.328 | 0.030 | 0.573 | **0.375** | 0.072 | **0.560** | 0.446 | 0.055 | 0.597 | 0.342 | 0.029 | 0.396 |
| CorrKG[†] | 0.404 | 0.071 | N/A | 0.365 | **0.045** | N/A | N/A | N/A | N/A | **0.449** | **0.079** | N/A | **0.359** | **0.044** | N/A |
| BART-large | 0.392 | 0.047 | 0.575 | 0.333 | 0.024 | 0.565 | 0.347 | 0.051 | 0.517 | 0.435 | 0.048 | 0.586 | 0.311 | 0.024 | 0.381 |
| SciBART-large | 0.396 | 0.057 | 0.587 | 0.328 | 0.026 | 0.557 | 0.329 | 0.056 | 0.503 | 0.421 | 0.050 | 0.567 | 0.304 | 0.033 | 0.382 |
| + TAPT | 0.426 | 0.063 | 0.597 | 0.330 | 0.030 | 0.569 | 0.347 | 0.064 | 0.519 | 0.442 | 0.055 | 0.585 | 0.333 | 0.031 | 0.386 |
| + TAPT + DESEL | **0.431** | **0.076** | **0.612** | **0.402** | 0.036 | **0.611** | 0.352 | **0.086** | 0.546 | **0.449** | 0.068 | **0.610** | 0.341 | 0.040 | **0.402** |

Table 3: Testing results on all datasets. $P$ and $A$ stand for $F1@M$ for present and absent keyphrases. $Sem$ stands for $SemF1$. The best performance is boldfaced. [†]copied from Zhao et al. (2022). The best entries in each column are statistically significantly higher than the second best (p < 0.05) via a paired bootstrap test. Full results in Table 8.

model or a newly trained "one2one" model[3] that learns to generate a single keyphrase based on $\mathcal{X}$. We use nucleus sampling with $p = 0.95$ and $\tau = 0.5$ for step 2, and set $m = 10$, $n = 10$, and $\alpha = 0.78$.

Table 3 presents the test results of important models in this paper. DESEL consistently improves the performance over the base model by a large margin. In Table 4, we compare against other selection strategies including random selection, input overlap using a sentence transformer model, and FreqFS proposed in Zhao et al. (2022). DESEL is the only method that consistently outperforms both greedy search and nucleus sampling.

**Discussion** Compared to the single-sequence decoding baselines, DESEL wins by bringing in the diversity. Compared to the baseline ranking methods, DESEL wins by capturing the correlations between labels (encoded in the greedy search outputs) and using the likelihood-based criteria to filter out

| Method | $P$ | $A$ | $Sem$ |
|---|---|---|---|
| Greedy search ($G$) | 0.426 | 0.063 | 0.597 |
| Nucleus sampling ($S$) | 0.385 | 0.074 | 0.599 |
| Random Selection | 0.385 | 0.072 | 0.596 |
| Input Overlap | 0.402 | 0.064 | 0.611 |
| FreqFS (Zhao et al., 2022) | 0.426 | 0.072 | 0.610 |
| DESEL (self) | 0.426 | 0.070 | 0.608 |
| DESEL (one2one) | **0.431** | **0.076** | **0.612** |

Table 4: A comparison across different decoding strategies. Methods below the dotted line merge $G$ with $S$.

high-quality phrases from the diverse candidates.

**Efficiency** DESEL harms the inference latency as it generates multiple sequences. To improve the efficiency, one optimization is reusing the encoder's outputs for all the decoding and scoring operations. We implemented this strategy and benchmarked it with the BART (base) model. DESEL with n = 10 (1 greedy search and 10 sampling sequences decoded) takes 3.8x time compared to greedy decoding.

---

[3]Starting from KeyBART, the one2one model can be efficiently trained. We provide more details in appendix F.

# 6 Related Work

**Keyphrase Generation** Meng et al. (2017) propose the task of Deep Keyphrase Generation and a strong baseline model CopyRNN. Later works improve the architecture by adding correlation constraints (Chen et al., 2018) and linguistic constraints (Zhao and Zhang, 2019), exploiting learning signals from titles (Ye and Wang, 2018; Chen et al., 2019b), and hierarchical modeling the phrases and words (Chen et al., 2020). Ye and Wang (2018) reformulate the problem as generating a sequence of keyphrases, while Ye et al. (2021) further uses a set generation formulation to remove the influence of difference target phrase ordering. Other works include incorporating reinforcement learning (Chan et al., 2019; Luo et al., 2021), GANs (Swaminathan et al., 2020), and unifying KPE with KPG (Chen et al., 2019a; Ahmad et al., 2021). Meng et al. (2021) conduct an empirical study on architecture, generalizability, phrase order, and decoding strategies, with the main focus on models trained from scratch instead of PLMs.

**PLMs for KPG** More recently, Wu et al. (2021), Chowdhury et al. (2022), Wu et al. (2022a), Gao et al. (2022), and Wu et al. (2022b) consider fine-tuning prefix-LMs or seq2seq PLMs for KPG. Kulkarni et al. (2022) use KPG as a pre-training task to learn strong BART-based representations. Zhao et al. (2022) adopt optimal transport for loss design and propose frequency-based filtering for decoding to improve BART-based KPG.

# 7 Conclusion

This paper systematically investigated model selection and decoding for building KPG models with seq2seq PLMs. Our analyses suggested much more nuanced patterns beyond the "conventional wisdom" assumed by the majority of current literature. Our novel decoding strategy, DESEL, significantly improved the performance of greedy search across multiple datasets. More broadly, this study underscores the distinct nature of the KPG task. One should not blindly transpose conclusions or assumptions from other text generation tasks. Instead, they warrant careful re-evaluation and empirical validation. Our work also opens up exciting directions for future work, with deep groundings in keyphrase literature. For instance, making KPG models more robust, interpreting a KPG model, and designing better decoding algorithms for KPG.

## Limitations

While our study sheds light on important aspects of keyphrase generation (KPG) models, several limitations present opportunities for future research.

First, our analysis focuses on model selection and decoding and thus uses default cross entropy loss and original training set without data augmentations. Investigating how the discussed design choices with more recent data augmentation (Ray Chowdhury et al., 2022; Garg et al., 2022) or training strategies (Zhao et al., 2022) is an important future study. In addition, the best approach to combine the conclusions reached in this paper on long input KPG (Garg et al., 2022) or KPG models trained with reinforcement learning (Chan et al., 2019; Luo et al., 2021) worth future study.

Second, while in-domain pre-training combined with task adaptation was found to enhance KPG performance, we did not fully investigate the underlying mechanisms leading to these improvements. Further research could explore the interplay between these two aspects and uncover more granular insights into how they improve KPG.

Finally, although we revealed a compromise between performance optimization and model robustness, we did not delve into designing new methods for improving the robustness of these models against perturbed inputs. Future research could further explore techniques to mitigate this trade-off, developing models that maintain high performance while being resistant to input perturbations.

## Ethics Statement

S2ORC and OAGKX are released under the Creative Commons By 4.0 License. We perform text cleaning and email/URL filtering on S2ORC to remove sensitive information, and we keep OAGKX as-is. We use the keyphrase benchmarking datasets distributed by the original authors. No additional preprocessing is performed before fine-tuning except lower-casing and tokenization. We do not re-distribute any of the datasets used in this work.

Potential risks of SciBART include accidental leakage of (1) sensitive personal information and (2) inaccurate factual information. For (1), we carefully preprocess the data in the preprocessing stage to remove personal information, including emails and URLs. However, we had difficulties desensitizing names and phone numbers in the text because they overlapped with the informative content. For (2), since SciBART is pre-trained on scientific pa-

pers, it may generate scientific-style statements that include inaccurate information. We encourage the potential users of SciBART not to rely fully on its outputs without verifying their correctness.

Pre-training SciBART and fine-tuning the large T5 models are computationally heavy, and we estimate the total $CO_2$ emission to be around 3000 kg using the calculation application provided by Lacoste et al. (2019). We will release the fine-tuned checkpoints and we document the hyperparameters in the appendix section D to help the community reduce the energy spent optimizing PLMs for KPG and other various NLP applications.

## Acknowledgments

The research is supported in part by Taboola, NSF CCF-2200274, and an Amazon AWS credit award. We thank the Taboola team for the helpful discussion. We also thank anonymous reviewers, Da Yin, Tanmay Parekh, and other members of the UCLA-NLP group for their valuable feedback.

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

<h1 style="text-align:center">Supplementary Material: Appendices</h1>

## A  Test Set Statistics

Table 5 summarizes the statistics of all testing datasets we use. We use the version distributed by Meng et al. (2017). In this distribution, SemEval's documents and keyphrases are already stemmed while the other datasets are not.

| Dataset | #Examples | #KP | %AKP | |KP| |
|---------|-----------|-----|------|------|
| KP20k   | 20000     | 5.3 | 37.1 | 2.0 |
| Inspec  | 500       | 9.8 | 26.4 | 2.5 |
| Krapivin| 400       | 5.9 | 44.3 | 2.2 |
| NUS     | 211       | 11.7| 45.6 | 2.2 |
| SemEval | 100       | 14.7| 57.4 | 2.4 |

Table 5: Test sets statistics. #KP, %AKP, and |KP| refers to the average number of keyphrases per document, the percentage of absent keyphrases, and the average number of words that each keyphrase contains.

## B  Attention heads as keyphrase extractors

We present the KPE performance of ranking noun phrase candidate using their attention intensities in Table 6. We observe that BART and T5 contain attention heads that function well as keyphrase extractors, some of which even surpass the KPE performance of the strong MPRank algorithm.

| Model | Size | Head | PKP $F1@5$ | $SemF1@5$ |
|-------|------|------|------------|-----------|
| MPRank | - | - | 0.188 | 0.429 |
| BART-base | 140M | 5-3 | 0.182 | 0.451 |
| BART-large | 406M | 11-9 | **0.191** | **0.452** |
| T5-small | 60M | 4-4 | 0.183 | 0.420 |
| T5-base | 223M | 8-1 | 0.164 | 0.388 |
| T5-large | 770M | 17-15 | 0.187 | 0.436 |
| T5-3B | 3B | 21-27 | 0.183 | 0.426 |

Table 6: Keyphrase extraction performance of utilizing attention intensity as ranking criteria. BART-large's attention head 11-9 is even able to outperform MPRank. PKP = present keyphrases.

## C  SciBART Pre-training Details

**Corpus and Data Preprocessing**  The S2ORC dataset contains over 100M papers from a variety of disciplines (Figure 3). We train on all the titles and abstracts to increase the coverage of different topics. After removing non-English[4] or title-only

entries, we fix wrong Unicode characters, remove emails and URLs, and convert the text to ASCII encoding[5]. The final dataset contains 171.7M documents or 15.4B tokens in total. We reserve 10k documents for validation and 10k for testing and use the rest as training data.

**Vocabulary**  Beltagy et al. (2019) suggest that using a domain-specific vocabulary is crucial to downstream in-domain fine-tuning performance. Following their observations, we build a cased BPE vocabulary in the scientific domain using the SentencePiece[6] library on the cleaned training data. We set the vocabulary size to 30K.

**Training**  For the pre-training objective, we only use text infilling as introduced in Lewis et al. (2020). We mask 30% of all tokens in each example, with the spans randomly sampled from a Poisson distribution ($\lambda = 3.5$). For 10% of the spans selected to mask, we replace them with a random token instead of the mask token. We set the maximum sequence length to 512. The model is pre-trained for 250k steps with batch size 2048, learning rate 3e-4, 10k warm-up steps, and polynomial learning rate decay. We use the Adam optimizer for pre-training. Using 8 Nvidia A100 GPUs (40G each), the training took eight days for SciBART-base and twelve days for SciBART-large.

## D  Implementation details

### D.1  Keyphrase Generation

For keyphrase generation with BART and T5, we use Huggingface Transformers and train for 15 epochs with early stopping. We use learning rate 6e-5, polynomial decay, batcsh size 64, and the AdamW optimizer. To fine-tune SciBART-base and SciBART-large, we use the Translation task provided by fairseq[7] and train for 10 epochs. We use learning rate 3e-5, polynomial decay, and the AdamW optimizer.

We perform a careful hyperparameter search over the learning rate, learning rate schedule, batch size, and warm-up steps. The corresponding search spaces are {1e-5, 3e-5, 6e-5, 1e-4, 3e-4}, {linear,

---

[4]We use guess_language for language detection.

[5]We use clean-text for data cleaning.

[6]https://github.com/google/sentencepiece

[7]https://github.com/facebookresearch/fairseq

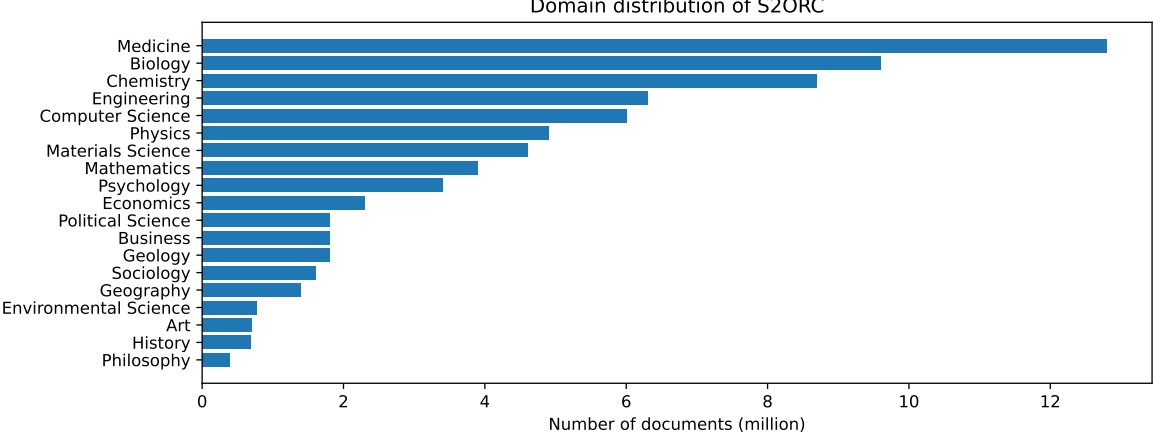

Figure 3: Domain distribution of the S2ORC dataset.

polynomial}, {16, 32, 64, 128}, and {500, 1000, 2000, 4000}. The best hyperparameters are presented in Table 7. For DESEL, we tune the hyperparameters on the validation set of KP20k. the search space for $\alpha$ is {0.5, 0.73, 0.78, 0.83, 0.88, 0.93, 0.98}.

The fine-tuning experiments are run on a local GPU server with RTX 2080 Ti (11G each) and A6000 GPUs (48G each). We use gradient accumulation to achieve the desired batch sizes.

## D.2 Baselines

For CopyTrans and SetTrans, we rerun with the original implementations to measure their performance. We use the earliest version of Key-BART available at https://zenodo.org/record/5784384#.Y0eToNLMJcA. For MPRank, we use the implementation from pke[8].

## E Prompting GPT-3.5 for paraphrasing

To generate the paraphrased titles and abstracts used in §4.4, we use the following prompt to query `gpt-3.5-turbo`:

```
Paraphrase the following title and
    abstract for a scientific paper.
    Your paraphrase should perfectly
    preserve the information in the
    original document and should be as
    formal as the original text. Do not
    change the spelling or case of the
    named entities in the original
    sentence. If the original entity is
    all lower-cased, do not upper-case
    the first letter of these names in
    your paraphrase.

Title: {original_title}
```

```
Abstract: {original_abstract}

Your Paraphrase:
Title:
```

We observe that the large language model can follow the specified format when generating the paraphrased titles and abstracts. During the postprocessing, we split the titles and the abstract from the response and directly use them for model testing. Figure 4 shows two examples of original and paraphrased text. The paraphrased text has high quality, preserves the scientific writing style, and retains most of the present keyphrases.

## F One2one model for DESEL

To better capture $\Pr(s_i|\mathcal{X})$ and $\Pr(g_j|\mathcal{X})$ for DE-SEL without the interference from other keyphrases co-occuring in the label, we propose to train an "one2one" model that learns to generate a single keyphrase given an input document. Starting from the KeyBART model, we fine-tune on the KP20k training set for 0.5 epoch with learning rate 5e-5, batch size 32, and the AdamW optimizer. We use a linear learning rate decay with 1000 warmup steps.

## G All model testing results

We present the testing scores as well as their standard deviation in Table 8. In addition to the models discussed in the main text, we also provide the performance of CatSeq (Yuan et al., 2020), CatSeqTG-2RF1 (Chan et al., 2019), ExHiRD-h (Chen et al., 2020) for further reference.

---

[8] https://github.com/boudinfl/pke

| Model | dropout | wdecay | optimizer | bsz | #epoch | #warm-up | lr | lr schedule |
|---|---|---|---|---|---|---|---|---|
| **Seq2seq PLMs** | | | | | | | | |
| T5-small | 0.1 | 0.01 | AdamW | 64 | 25 | 2000 | 6e-5 | polynomial |
| T5-base | 0.1 | 0.01 | AdamW | 64 | 15 | 2000 | 6e-5 | polynomial |
| T5-large | 0.1 | 0.01 | AdamW | 64 | 5 | 1000 | 6e-5 | polynomial |
| T5-3B | 0.1 | 0.01 | AdamW | 64 | 5 | 1000 | 6e-5 | polynomial |
| BART-base | 0.1 | 0.01 | AdamW | 64 | 15 | 2000 | 6e-5 | polynomial |
| BART-large | 0.1 | 0.01 | AdamW | 64 | 10 | 2000 | 6e-5 | polynomial |
| **In-domain Seq2seq PLMs** | | | | | | | | |
| SciBART-base | 0.1 | 0.01 | AdamW | 32 | 10 | 2000 | 3e-5 | polynomial |
| SciBART-large | 0.1 | 0.01 | AdamW | 64 | 10 | 2000 | 3e-5 | polynomial |
| **Task-adaptive Seq2seq Models** | | | | | | | | |
| FLAN-T5-small | 0.1 | 0.01 | AdamW | 64 | 25 | 2000 | 6e-5 | polynomial |
| FLAN-T5-base | 0.1 | 0.01 | AdamW | 64 | 15 | 2000 | 6e-5 | polynomial |
| FLAN-T5-large | 0.1 | 0.01 | AdamW | 64 | 5 | 1000 | 6e-5 | polynomial |
| FLAN-T5-XL | 0.1 | 0.01 | AdamW | 64 | 5 | 1000 | 6e-5 | polynomial |
| KeyBART | 0.1 | 0.01 | AdamW | 64 | 15 | 2000 | 3e-5 | polynomial |
| SciBART-base + OAGKX | 0.1 | 0.01 | AdamW | 64 | 5 | 1000 | 1e-5 | polynomial |
| SciBART-large + OAGKX | 0.1 | 0.01 | AdamW | 64 | 5 | 1000 | 1e-5 | polynomial |

Table 7: Hyperparameters for fine-tuning PLMs for keyphrase generation on KP20k. The hyperparameters are determined using the loss on the KP20k validation dataset. "wdecay" = weight decay, "bsz" = batch size, "#warm-up" = the number of warm-up steps, "lr" = learning rate, "lr schedule" = learning rate decay schedule. We use early stopping for all the models and use the model with the lowest validation loss as the final model.

---

**Original title:** hybrid analytical modeling of pending cache hits , data prefetching , and mshrs .
**Original abstract:** this article proposes techniques to predict the performance impact of pending cache hits , hardware prefetching , and miss status holding register resources on superscalar microprocessors using hybrid analytical models . the proposed models focus on timeliness of pending hits and prefetches and account for a limited number of mshrs . they improve modeling accuracy of pending hits by 3.9 x and when modeling data prefetching , a limited number of mshrs , or both , these techniques result in average errors of 9.5 % to 17.8 % . the impact of non uniform dram memory latency is shown to be approximated well by using a moving average of memory access latency .
**Paraphrased title:** a hybrid analytical model for pending cache hits , data prefetching , and mshrs .
**Paraphrased abstract:** this scientific paper presents a novel approach to predicting the performance impact of pending cache hits , hardware prefetching , and miss status holding register resources on superscalar microprocessors . the proposed hybrid analytical models focus on the timeliness of pending hits and prefetches , while also accounting for a limited number of mshrs . by utilizing these models , the accuracy of pending hit modeling is improved by 3.9 times . when modeling data prefetching , a limited number of mshrs , or both , these techniques result in average errors ranging from 9.5 % to 17.8 % . additionally , the paper demonstrates that the impact of non - uniform dram memory latency can be approximated well by using a moving average of memory access latency .

**Original title:** a variant of parallel plane sweep algorithm for multicore systems .
**Original abstract:** parallel algorithms used in very large scale integration physical design bring significant challenges for their efficient and effective design and implementation . the rectangle intersection problem is a subset of the plane sweep problem , a topic of computational geometry and a component in design rule checking , parasitic resistance capacitance extraction , and mask processing flows . a variant of a plane sweep algorithm that is embarrassingly parallel and therefore easily scalable on multicore machines and clusters , while exceeding the best known parallel plane sweep algorithms on real world tests , is presented in this letter .
**Paraphrased title:** a modified parallel plane sweep algorithm for multicore systems .
**Paraphrased abstract:** the design and implementation of parallel algorithms for physical design in very large scale integration pose significant challenges . the rectangle intersection problem , which is a component in design rule checking , parasitic resistance capacitance extraction , and mask processing flows , is a subset of the plane sweep problem in computational geometry . this letter presents a variant of the plane sweep algorithm that is embarrassingly parallel and can be easily scaled on multicore machines and clusters . the proposed algorithm outperforms the best known parallel plane sweep algorithms on real - world tests .

Figure 4: Examples of documents paraphrased by `gpt-3.5-turbo`. We color the present keyphrases in blue.

| Method | \|M\| | KP20k | | | Inspec | | | Krapivin | | | NUS | | | SemEval | | |
|---|---|---|---|---|---|---|---|---|---|---|---|---|---|---|---|---|
| | | $P$ | $A$ | $Sem$ | $P$ | $A$ | $Sem$ | $P$ | $A$ | $Sem$ | $P$ | $A$ | $Sem$ | $P$ | $A$ | $Sem$ |
| **Trained from-scratch architectures** | | | | | | | | | | | | | | | | |
| CatSeq | 21M | 0.367 | 0.032 | N/A | 0.262 | 0.008 | N/A | 0.354 | 0.036 | N/A | 0.397 | 0.028 | N/A | 0.283 | 0.028 | N/A |
| ExHiRD-h | 22M | $0.374_0$ | $0.025_0$ | N/A | $0.291_3$ | $0.016_2$ | N/A | $0.308_4$ | $0.033_4$ | N/A | N/A | N/A | N/A | $0.282_{18}$ | $0.021_6$ | N/A |
| CopyTrans | 98M | $0.376_2$ | $0.046_4$ | $0.562_1$ | $0.333_5$ | $0.023_1$ | $0.569_4$ | $0.365_7$ | $0.063_4$ | $0.547_3$ | $0.429_9$ | $0.044_9$ | $0.579_2$ | $0.321_8$ | $0.022_4$ | $0.377_1$ |
| SetTrans | 98M | $0.391_2$ | $0.058_1$ | $0.585_2$ | $0.328_1$ | $0.030_1$ | $0.573_1$ | $0.375_{11}$ | $0.072_3$ | $0.560_0$ | $0.446_{22}$ | $0.055_{17}$ | $0.597_1$ | $0.342_{14}$ | $0.029_2$ | $0.396_3$ |
| **PLM-based methods** | | | | | | | | | | | | | | | | |
| CorrKG$^\dagger$ | 140M | 0.404 | 0.071 | N/A | 0.365 | 0.045 | N/A | N/A | N/A | N/A | 0.449 | 0.079 | N/A | 0.359 | 0.044 | N/A |
| BART-base | 140M | $0.388_3$ | $0.042_2$ | $0.571_2$ | $0.323_7$ | $0.017_2$ | $0.561_5$ | $0.336_6$ | $0.049_6$ | $0.514_6$ | $0.424_8$ | $0.042_9$ | $0.581_3$ | $0.321_{21}$ | $0.021_2$ | $0.372_2$ |
| BART-large | 406M | $0.392_2$ | $0.047_2$ | $0.575_1$ | $0.333_9$ | $0.024_4$ | $0.565_6$ | $0.347_3$ | $0.051_2$ | $0.517_5$ | $0.435_{11}$ | $0.048_9$ | $0.586_7$ | $0.311_{16}$ | $0.024_3$ | $0.381_6$ |
| KeyBART | 406M | $0.398_2$ | $0.047_1$ | $0.576_1$ | $0.325_5$ | $0.023_2$ | $0.561_4$ | $0.365_{14}$ | $0.064_6$ | $0.533_2$ | $0.430_{10}$ | $0.055_7$ | $0.582_2$ | $0.289_4$ | $0.022_5$ | $0.365_4$ |
| SciBART-base | 124M | $0.396_2$ | $0.052_4$ | $0.576_5$ | $0.328_8$ | $0.028_4$ | $0.562_4$ | $0.329_{11}$ | $0.054_8$ | $0.502_8$ | $0.421_{14}$ | $0.053_2$ | $0.572_8$ | $0.304_8$ | $0.022_1$ | $0.376_4$ |
| + TAPT | 124M | $0.415_2$ | $0.052_1$ | $0.590_1$ | $0.330_6$ | $0.027_4$ | $0.573_2$ | $0.337_9$ | $0.057_7$ | $0.514_7$ | $0.424_6$ | $0.048_2$ | $0.579_2$ | $0.329_9$ | $0.024_0$ | $0.388_5$ |
| SciBART-large | 386M | $0.396_4$ | $0.057_3$ | $0.587_2$ | $0.328_{13}$ | $0.026_2$ | $0.557_4$ | $0.329_{12}$ | $0.056_3$ | $0.503_7$ | $0.421_{12}$ | $0.050_7$ | $0.567_9$ | $0.304_{12}$ | $0.033_8$ | $0.382_9$ |
| + TAPT | 386M | $0.426_0$ | $0.063_1$ | $0.597_1$ | $0.330_4$ | $0.030_1$ | $0.568_4$ | $0.347_6$ | $0.064_7$ | $0.520_{10}$ | $0.442_{11}$ | $0.055_5$ | $0.585_5$ | $0.333_{19}$ | $0.031_2$ | $0.386_9$ |
| T5-small | 60M | $0.346_0$ | $0.019_0$ | $0.543_0$ | $0.354_3$ | $0.026_2$ | $0.572_2$ | $0.298_2$ | $0.026_1$ | $0.495_1$ | $0.398_0$ | $0.020_4$ | $0.562_2$ | $0.315_7$ | $0.013_1$ | $0.377_1$ |
| T5-base | 223M | $0.388_0$ | $0.034_0$ | $0.570_0$ | $0.339_4$ | $0.020_3$ | $0.574_2$ | $0.350_1$ | $0.043_3$ | $0.524_1$ | $0.440_4$ | $0.051_2$ | $0.589_1$ | $0.326_{13}$ | $0.020_4$ | $0.383_3$ |
| T5-large | 770M | $0.393_0$ | $0.035_0$ | $0.573_0$ | $0.343_3$ | $0.021_5$ | $0.573_1$ | $0.359_4$ | $0.045_6$ | $0.536_3$ | $0.438_5$ | $0.042_5$ | $0.587_5$ | $0.321_9$ | $0.020_2$ | $0.391_3$ |
| T5-3B | 3B | $0.419_1$ | $0.046_1$ | $0.587_0$ | $0.319_5$ | $0.021_2$ | $0.559_1$ | $0.353_6$ | $0.048_1$ | $0.526_1$ | $0.460_{10}$ | $0.050_8$ | $0.600_4$ | $0.337_7$ | $0.027_4$ | $0.391_3$ |
| FLAN-T5-small | 60M | $0.362_0$ | $0.024_0$ | $0.552_0$ | $0.354_1$ | $0.018_2$ | $0.569_0$ | $0.313_3$ | $0.024_0$ | $0.498_1$ | $0.398_1$ | $0.030_4$ | $0.568_1$ | $0.321_2$ | $0.017_1$ | $0.385_1$ |
| FLAN-T5-base | 223M | $0.393_1$ | $0.035_0$ | $0.574_0$ | $0.358_4$ | $0.020_3$ | $0.577_4$ | $0.340_3$ | $0.049_4$ | $0.514_1$ | $0.436_0$ | $0.029_2$ | $0.586_0$ | $0.333_4$ | $0.017_2$ | $0.387_0$ |
| FLAN-T5-large | 770M | $0.404_2$ | $0.036_0$ | $0.579_0$ | $0.347_1$ | $0.018_4$ | $0.573_1$ | $0.347_6$ | $0.041_2$ | $0.526_4$ | $0.441_4$ | $0.046_6$ | $0.598_0$ | $0.327_{18}$ | $0.021_2$ | $0.390_6$ |
| FLAN-T5-XL | 3B | $0.427_1$ | $0.049_2$ | $0.592_1$ | $0.313_2$ | $0.024_1$ | $0.556_2$ | $0.356_1$ | $0.047_3$ | $0.517_4$ | $0.455_4$ | $0.054_2$ | $0.595_2$ | $0.332_5$ | $0.025_1$ | $0.392_2$ |

Table 8: Testing results on all five datasets. $P$ and $A$ stand for $F1@M$ for present keyphrases and absent keyphrases. $Sem$ stands for $SemF1$. The reported results are averaged across three runs with different random seeds. The standard deviation of each entry is presented in the subscript. For example, $23.5_6$ means an average of 23.5 with a standard deviation of 0.6. We omit the subscript for methods with a single run. $^\dagger$copied from Zhao et al. (2022).