# OpenReview forum: "Rethinking Model Selection and Decoding for Keyphrase Generation with Pre-trained Sequence-to-Sequence Models"
_EMNLP/2023/Conference — EMNLP 2023 Main_

### Official Review · Reviewer_vDDj · 2023-08-04

**Soundness:** 4

**Excitement:**

3: Ambivalent: It has merits (e.g., it reports state-of-the-art results, the idea is nice), but there are key weaknesses (e.g., it describes incremental work), and it can significantly benefit from another round of revision. However, I won't object to accepting it if my co-reviewers champion it.

**Paper Topic And Main Contributions:**

This paper presents systematic experiments on PLM choice, pre-training corpus, pre-training tasks, and decoding strategies in Keyphrase Generation and provides corresponding explanations and analyses. The motivation is that the current work often overlooks the choices of models, leading to KPG systems being compared under different backbones, which is not conducive to effectively comparing the effects of various designs.

**Questions For The Authors:**

1.	Does the DESEL decoding strategy designed in this paper reduce the efficiency of inference? Has the author thoroughly explored this aspect?
2.	How should one select the appropriate DESEL decoding hyperparameter α setting? There is a lack of relevant hyperparameter setting experiments in the paper.


**Reasons To Accept:**

1. This paper showcases thorough experiment on multiple datasets and various backbones. The experiment design is comprehensive, demonstrating a solid workload.
2. The decoding strategy devised in this study is both straightforward and powerful, leading to significant improvements across multiple data sets.
3. The paper is easy to follow and the question explored is of great important.


**Reasons To Reject:**

1.	The baseline for comparison is too limited and a little old. Consider incorporating more strong baselines for a comprehensive evaluation.
2.	The parameter scale of the selected backbone is relatively small. Hence, it raises questions about the existence of parameter inefficient on a larger scale model. And are the relevant experimental results still valid under these conditions?
3.	There have been no tests to determine the statistical significance of the results.


**Reproducibility:**

3: Could reproduce the results with some difficulty. The settings of parameters are underspecified or subjectively determined; the training/evaluation data are not widely available.

**Reviewer Confidence:**

4: Quite sure. I tried to check the important points carefully. It's unlikely, though conceivable, that I missed something that should affect my ratings.

---

> ### Author Rebuttal · Authors · 2023-08-28
>
> Thank you for the insightful review.
>
> (1) The baseline is limited and a little old.
>
> In our preliminary studies, we also considered baselines beyond SetTrans, KeyBART, and CorrKG: AutoKeyGen [1], SEG-Net [2], CatSeqTG-2RF1 [3], ExHiRD-h [4], and UniKeyphrase [5]. However, we find that the non-PLM baselines have weaker results than SetTrans, and that UniKeyphrase is weaker than CorrKG. As the goal of this work is thoroughly studying PLM-based methods, we only provide the results of the strongest baselines as a point of reference. In our final revision, we will include the results of [1] - [5] in the appendix.
>
> (2) Scaling the results to larger models.
>
> We did not consider models larger than 3B due to a limited computation budget. We note that in KPG applications, the set of desired keyphrases often changes quickly and the model may need to be frequently updated. Therefore, it is usually infeasible to fully fine-tune and serve models much larger than 3B. For these models, prompting may be a more feasible solution. However, a more fine-grained evaluation protocol may be required to assess their ability as it is unfair to directly compare with models trained with large supervised datasets [6, 7]. Therefore, we leave the investigation on them to future work.
>
> (3) Statistical significance.
>
> Thank you for the suggestion. We have performed paired bootstrap testing for the results reported in the main text. For Table 3, the best entries in each column are statistically significant than the second best (p < 0.05) except the “P” column in NUS and the “Sem” column in SemEval. In Table 1 and Table 4, the best entries in each column are statistically significant than the second best (p < 0.05). We will add the statistical testing results in the final revision.
>
> (4) Inference efficiency of DESEL.
>
> We acknowledge that DESEL will harm the inference latency because it generates multiple sequences. To reduce the latency, a simple optimization is caching and reusing the encoder’s outputs for all the decoding and scoring operations. We implemented this strategy and benchmarked with BART-base. DESEL with n = 10 (1 greedy search and 10 sampling sequences decoded) takes 3.8x time compared to greedy decoding. We will discuss the latency issue in our final revision.
>
> (5) How to tune α for DESEL?
>
> We tuned on the KP20k validation set and used the same α=0.78 for all the testing datasets. The search space is {0.5, 0.73, 0.78, 0.83, 0.88, 0.93, 0.98}. We will add this to our final revision.
>
>
> References
>
> [1] Unsupervised Deep Keyphrase Generation (Shen et al., AAAI 2022)
>
> [2] Select, Extract and Generate: Neural Keyphrase Generation with Layer-wise Coverage Attention (Ahmad et al., ACL-IJCNLP 2021)
>
> [3] Neural keyphrase generation via reinforcement learning with adaptive rewards (Chan et al., ACL 2019)
>
> [4] Exclusive Hierarchical Decoding for Deep Keyphrase Generation (Chen et al., ACL 2020)
>
> [5] UniKeyphrase: A Unified Extraction and Generation Framework for Keyphrase Prediction (Wu et al., Findings 2021)
>
> [6] KPEval: Towards Fine-grained Semantic-based Evaluation of Keyphrase Extraction and Generation Systems (Wu et al., arXiv 2023)
>
> [7] Is ChatGPT A Good Keyphrase Generator? A Preliminary Study (Song et al., arXiv 2023)

---

### Official Review · Reviewer_3t8S · 2023-08-05

**Soundness:** 4

**Excitement:**

3: Ambivalent: It has merits (e.g., it reports state-of-the-art results, the idea is nice), but there are key weaknesses (e.g., it describes incremental work), and it can significantly benefit from another round of revision. However, I won't object to accepting it if my co-reviewers champion it.

**Paper Topic And Main Contributions:**

This paper reviews the Keyphrase Generation (KG, or named KPG in this paper) task and concludes a better model selection strategy and decoding strategy for the KG task. From the perspective of model selection, this paper first provides some empirical results about the PLM selection and then explores the influence of in-domain pre-training and task-adaptive pre-training. From the perspective of decoding strategy, this paper compares six decoding algorithms and proposes DESEL which includes the decoding, sampling, and selection stages.

**Reasons To Accept:**

1. This paper reviews the PLM-based KG methods and focuses on the decoding stage, which is exciting and lacks exploration.
2. This paper summarizes and empirically explores the factors influencing generation performance.
3. DESEL, the proposed probability-based decode-select algorithm, can improve the greedy search.

**Reasons To Reject:**

1. It would be better if the authors could discuss in depth why DESEL improves performance compared with other decoding methods.
2. Reinforcement learning has been proven effective in RNN-based KG methods, but this factor is not considered in this revisiting work.

**Reproducibility:**

3: Could reproduce the results with some difficulty. The settings of parameters are underspecified or subjectively determined; the training/evaluation data are not widely available.

**Reviewer Confidence:**

4: Quite sure. I tried to check the important points carefully. It's unlikely, though conceivable, that I missed something that should affect my ratings.

---

> ### Author Rebuttal · Authors · 2023-08-28
>
> Thank you for the insightful review.
>
> (1) Why DESEL improves performance compared with other decoding methods?
>
> The success of DESEL is based on two insights collected from the empirical study in section 5.1: 1) greedy search produces high quality keyphrases and 2) path dependency reduces the diversity of greedy search, which is less of an issue for other decoding methods. Therefore, DESEL adopts the novel formulation of selecting high quality phrases from a diverse set of candidates to complement greedy search. Compared to the single-sequence decoding baselines, DESEL wins by bringing in the diversity. Compared to the ranking-based baselines (such as using ranking-based KPE methods to rerank KPG models’ outputs), DESEL wins by capturing the correlations between labels (encoded in the greedy search outputs) and using the probability-based formulation to filter out high quality phrases from the diverse candidates. We will further explain this point with qualitative examples in the final revision.
>
> (2) RL is not considered in this work.
>
> We note that RL is an orthogonal factor to the dimensions considered in this work. Based on the existing works that report good performance on using RL for KPG [1, 2, 3], it is usually assumed that a supervised fine-tuned KPG model is required before starting RL training. However, how to obtain a strong PLM-based supervised model is not yet clear. We bridge this gap in this paper and delegate combining the proposed approaches with RL to future work. We will acknowledge this point in the limitation section.
>
> References
>
> [1] Neural keyphrase generation via reinforcement learning with adaptive rewards (Chan et al., ACL 2019)
>
> [2] Keyphrase Generation with GANs in Low-Resources Scenarios (Lancioni et al., sustainlp 2020)
>
> [3] Keyphrase Generation with Fine-Grained Evaluation-Guided Reinforcement Learning (Luo et al., Findings 2021)

---

### Official Review · Reviewer_TvwR · 2023-08-07

**Soundness:** 3

**Excitement:**

3: Ambivalent: It has merits (e.g., it reports state-of-the-art results, the idea is nice), but there are key weaknesses (e.g., it describes incremental work), and it can significantly benefit from another round of revision. However, I won't object to accepting it if my co-reviewers champion it.

**Paper Topic And Main Contributions:**

This paper proposes a new keyphrase generation (KPG) approach. Basically, the authors analyzed previous work by posing several hypotheses and then proposed an decode selection algorithm called DESE improves greedy search output considerably. The proposed approach achieved state-of-the-art results in several benchmarks.

**Questions For The Authors:**

Question A:
SemEval, Krapivin and NUS datasets contains full texts apart from abstracts. The paper does not mention anywhere whether the full texts was used as input or only abstracts. Can you clarify which one?

Question B:
If the answer of Question A is abstracts only, then how do you envisage your proposed DESEL approach will work when the input is full text (i.e. text consisting of several thousand words)?

**Reasons To Accept:**

1. Various analysis of the impact of existing design choices for KPG solutions
2. Proposed approach achieved state-of-the-art results in several benchmarks.

**Reasons To Reject:**

1. The proposed supervised approach will work for input small text (e.g. abstracts). But it is not obvious how this will scale for large text (i.e. full documents) where existing unsupervised approaches (e.g. AutoKeyGen (Shen et al, AAAI 2022) will likely do a better job with respect to the proposed approach.
2. The writing and organization of the paper could be improved. The authors frequently referred to the Appendixes inside the main body (8 pages) of the paper for various optional things (e.g. analysis of MPRank) but could have used such spaces in the main body to discuss important things such as what limitations their approach have w.r.t. the length of the input text.

**Reproducibility:**

3: Could reproduce the results with some difficulty. The settings of parameters are underspecified or subjectively determined; the training/evaluation data are not widely available.

**Reviewer Confidence:**

3: Pretty sure, but there's a chance I missed something. Although I have a good feel for this area in general, I did not carefully check the paper's details, e.g., the math, experimental design, or novelty.

---

> ### Author Rebuttal · Authors · 2023-08-28
>
> Thank you for the insightful review.
>
> (1) How can the proposed approach scale for long input text?
>
> We argue that scaling the approach to longer sequences is natural and in fact explored by previous work. Specifically, efficient attention has been shown effective in fine-tuning BART-like PLMs [1, 2]. In addition, the PLM can be trained to first compress the context and then generate the keyphrases [3]. The first section of the paper complements the long document fine-tuning approaches by exploring understudied factors such as model size, objective, or training data, providing useful guidance on selecting the best PLM to fine-tune.
>
> In addition, it is unclear whether AutoKeyGen will be a better candidate with long document inputs. The main reason is that AutoKeyGen uses BiLSTM as the encoder, which may not handle long sequences as well as sparse transformers [1, 4]. There is also no existing work showing the AutoKeyGen can excel in the existing long document KPG datasets.
>
> (2) Paper organization.
>
> Thank you for the suggestions. In the final revision, we will keep more relevant details within the main text and reserve space for a detailed discussion of the limitations of this study.
>
> (3) Which version of the datasets do you test on?
>
> As this paper already comprehensively explores various settings, we follow the established KPG literature to restrict the scope of the study to extracting keyphrases from titles and abstracts [5, 6, 7, 8]. We leave the study for long documents to future work.
>
> (4) How can DESEL scale for long input text?
>
> DESEL can operate on any sequence-to-sequence KPG models and it is agnostic to the input size. To employ DESEL on long sequences, the main challenge is on the KPG model itself. As mentioned in (1), we would expect the best performance to be achieved by using DESEL with a BART model fine-tuned with efficient attentions on long-document KPG [2].
>
> References
>
> [1] Longformer: The Long-Document Transformer (Beltagy et al., arXiv 2020)
>
> [2] Keyphrase Generation Beyond the Boundaries of Title and Abstract (Garg et al., Findings 2022)
>
> [3] Select, Extract and Generate: Neural Keyphrase Generation with Layer-wise Coverage Attention (Ahmad et al., ACL-IJCNLP 2021)
>
> [4] Generating long sequences with sparse transformers (Child et al., arXiv 2019)
>
> [5] Deep Keyphrase Generation (Meng et al., ACL 2017)
>
> [6] Neural keyphrase generation via reinforcement learning with adaptive rewards (Chan et al., ACL 2019)
>
> [7] One Size Does Not Fit All: Generating and Evaluating Variable Number of Keyphrases (Yuan et al., ACL 2020)
>
> [8] ONE2SET: Generating Diverse Keyphrases as a Set (Ye et al., ACL 2021)

---

### Official Review · Reviewer_FavJ · 2023-08-11

**Soundness:** 4

**Excitement:**

4: Strong: This paper deepens the understanding of some phenomenon or lowers the barriers to an existing research direction.

**Paper Topic And Main Contributions:**

Paper topic: Keyphrase Generation, performance evaluation

Main contributions:
1. The paper systematically studied scaling up the model size, in-domain pre-training, and task adaptation and show that only certain combinations are effective in improving the KPG performance, with a compromise on robustness.
2. The auhtors established the trade-off between accuracy and concept coverage for different decoding algorithms. Then, we introduce a probability-based decode-select mechanism DESEL that consistently improves over greedy search.
3. The study sheds light on how previously under-explored factors can have profound influence on keyphrase generation performance.

**Questions For The Authors:**

The lower F1-scores in case of "absent keyphrases". Have you investigated the reasons behind this?

**Reasons To Accept:**

1. The paper is well written and well organized.
2. The authors provided detailed analysis and ablation study to claim their contribution.
3. The paper tried to follow a clear reasoning behind the experimental results.
4. The findings of the paper (in domain pre-training with task adaptive pre-training and proposed decoding strategy) will help the community in keyphrase generation task.

**Reasons To Reject:**

The reasons behind the improvements of in domain pre-training+task adaptive pre-training is not clearly discussed. The experimental results showed improvement in all cases (present keyphrase, absent keyphrase) only for the KP20k dataset. However, in case of other datasets, the F1-score for absent keyphrase is lower than the traditional approaches.

**Reproducibility:**

4: Could mostly reproduce the results, but there may be some variation because of sample variance or minor variations in their interpretation of the protocol or method.

**Reviewer Confidence:**

5: Positive that my evaluation is correct. I read the paper very carefully and I am very familiar with related work.

**Typos Grammar Style And Presentation Improvements:**

The line at 1153, "We observe that the large language model can following the specified format when generating the paraphrased titles and abstracts." is not fully complete.

---

> ### Author Rebuttal · Authors · 2023-08-28
>
> Thank you for the insightful review.
>
> (1) The performance gain is inconsistent. Absent keyphrase performance is low.
>
> We first note that the score improvement trends of present and absent keyphrases are consistent across the studied PLM settings (Figure 1) and datasets (last four rows of Table 3). In most of the cases (P, A, and Sem) SciBART+TAPT is better than BART or SciBART, and in all the cases, SciBART+TAPT+DECEL is the best among the four PLM settings. Indeed, our results show that SciBART+TAPT cannot outperform SetTrans. We hypothesize the difference comes from SetTrans’ novel set loss formulation, which enhances the training and decoding efficiency. On the other hand, SciBART+TAPT is limited by MLE training and greedy decoding. To bridge the gap, we introduce DESEL which significantly boosts the decoding performance of SciBART+TAPT.
>
> We also note that it is expected that the results may not be state-of-the-art compared to CorrKG, as CorrKG introduces an effective loss term to boost the performance of BART. This is complementary to the main argument of the paper, that only considering the best model and decoding algorithm choices would bring significant performance gains without the optimizations made e.g., in CorrKG.
>
> (2) Typo.
>
> Thank you for pointing this out. We will proofread and improve the writing in the final version.

---

### Meta-Review · Area_Chair_bFab · 2023-09-27

**Recommendation:** 3

**Metareview:**

Although the reviewers overall acknowledge several contributions of the paper, such as 1) detailed experiments and ablation study,  2) improvements over the baselines, and 3) analyses of under-explored factors that influence keyphrase generation performance, there are concerns such as the scalability of the method to longer texts, as well as further clarification on the results obtained.

---

### Decision · Program_Chairs · 2023-10-07

**Decision:**

Accept-Main

**Comment:**

Although the reviewers overall acknowledge several contributions of the paper, such as 1) detailed experiments and ablation study,  2) improvements over the baselines, and 3) analyses of under-explored factors that influence keyphrase generation performance, there are concerns such as the scalability of the method to longer texts, as well as further clarification on the results obtained.